# Universal Algorithm for Extreme Bandits with the Minimal Complexities

## Abstract

The Multi-Armed Bandit is a classic reinforcement learning problem that exemplifies the exploration–exploitation trade-off dilemma. When extreme values rather than expected values are of interest, the Extreme Bandit is introduced. The motivation for this work comes from black-box optimization problems and meta learning, where the goal is to find the best value for a target function from different search spaces or using multiple search heuristics. Previous work on the extreme bandit problem has assumed that rewards are drawn from an i.i.d manner, which severely limits the applicability of this class of algorithm. In this paper, with minimal temporal and spatial cost and minimal assumptions about the reward distribution, we present an novel algorithm and provide its analysis. Numerical experiments highlight the performance of the proposed algorithm to the existing approaches.

## 1 Introduction

The Multi-Armed Bandit (MAB) often serves as a powerful analogy for balancing exploration and exploitation in search domains or sequential decision making tasks. In a MAB model, the learner is interacting with $K$ arms, each arm generates rewards from a different distribution, which is not known a-priori. The problem is to allocate trials to the arms with the goal of maximizing cumulative reward, and has been widely applied in diverse domains from real-life online recommending systems (Hill et al., 2017) and Go games (Silver et al., 2016) to various optimization or selection problems in machine learning research (Nguyen et al., 2020) (Bouneffouf et al., 2020).

In this paper, we consider a variant of the MAB, called the Extreme Bandits. The problem is to allocate trials among the $K$ arms with the goal of maximizing the expected best single reward. Our work is motivated by black-box optimization problems for which a number of search heuristics exist: given $K$ heuristics, e.g. Bayesian Optimization procedures in $K$ independent sub-spaces, each yielding unknown outcomes sequentially when applied to some particular search space, we wish to find a single best value for the target black-box function, as in many meta learning problems. Similar interest is discussed in Nishihara et al. (2016): Given the wide variety of hyper-parameter optimization (HPO) algorithms available, it's beneficial and challenging to devise a strategy for selecting which algorithms to use in a sequential order so that the general performance is close to what it would have been if only one of the best algorithms was used. This setting is also natural in more real-world scenarios such as project scheduling problems (Cicirello & Smith, 2005), malicious client detection in federated learning systems, and anomaly detection in endhost traffic models (Bhatt et al., 2023).

In addition to considering its applications, the extreme bandit framework has distinct challenges that deserve individual focus. For example, different from the classical MAB, the "best" distribution in extreme bandit depends on the rewards already obtained and on the remaining time span. Consequently, no policy can be ensured to perform as well as an oracle that always selects the best option over a specific period (Nishihara et al., 2016).

### 1.1 Main contributions

In this work, we revisit the extreme bandit problem with the idea of designing algorithms with minimal temporal and spatial cost under minimal assumptions on the arms. Our main contributions are as follows:

- We provided a universal extreme bandit algorithm *A\* Extreme Bandit* (`AEB`) for $K$-armed bandits that has $\mathcal{O}(T \log K)$ time complexity, which is the lowest in non-ETC (Explore-Then-Commit) algorithms within our knowledge, and $\mathcal{O}(K)$ storage complexity independent of the time horizon $T$, which is the lowest reported in the existing literature (Section 5.1). The algorithm is fashioned in a non-parametric way by considering only the number of trails and the observed maximum elements of each arm.

- We construct a unique surrogate of the bandit feedback value to trade-off exploration and exploitation (Section 3). Leveraging similar analysis scheme to the classic A\* (A-star) search algorithm, `AEB` is shown asymptotically optimal, i.e, with probability 1 the best arm will have the most allocated trials eventually (Section 4).

- To the best of our knowledge, `AEB` is the first algorithm in the extreme bandit literature with a performance guarantee when the rewards are not generated independently and identically distributed (i.i.d), as in many HPO algorithms. Besides, `AEB` is distribution-free, i.e., it doesn't make any parametric assumptions on the distributions of the reward.

- Although with the weakest assumptions introduced so far for extreme bandits, `AEB` achieves competitive performance in common reported parametric arms such as polynomial and exponential tails, and state-of-the-art performance in more realistic problems with non-i.i.d or non-stationary arms (Section 5).

## 1.2 RELATED WORK

Following the criterion proposed by Bhatt et al. (2023) and Baudry et al. (2022), existing algorithms for extreme bandit can be categorized into three:

- Fully-parametric methods (Cicirello & Smith, 2005) (Streeter & Smith, 2006a) assume the arms are drawn from known distributions, two types of generalized extreme value distribution (GEV distribution Type I - Gumbel & Type II - Frechet) are discussed.

- Semi-parametric methods have weaker semi-parametric assumptions on the distributions of the rewards are drawn from. Assuming that a lower bound on a parameter of the distribution is known, Carpentier & Valko (2014) and Achab et al. (2017) obtain vanishing regrets (Section 2) for second-order Pareto distributions. David & Shimkin (2016) assumes a known lower bound on the tail distribution.

- Distribution-free methods make no assumptions on the family of the rewards distribution. Inspired by Chernoff Interval Estimation applied in the Upper Confidence Bound algorithm (Auer et al., 2002) for classical MAB, Streeter & Smith (2006b) proposed a simple algorithm called ThresholdAscent, but without theoretical guarantees. Bhatt et al. (2023) proposed Max-Median by considering the median of maximum elements of carefully designed sub-sets of observed data. Max-Median can be employed for any kind of i.i.d distribution, and its vanishing regrets for polynomial and exponential tails are proved. Similar in spirit to Max-Median, Baudry et al. (2022) recently introduced Quantile of Maxima algorithm with two strategies: explore-then-commit style QoMax-ETC, and QoMax-SDA supported by a recent sub-sampling method. Both strategies are shown to have vanishing regrets and empirically more efficient than prior approaches.

## 2 EXTREME BANDIT SETTING

We denote the $K$ arms generating rewards by $\nu_1, \ldots, \nu_K$. Let $X_{k,t}$ be the reward obtained from arm $\nu_k$ at time $t$, the rewards stream $(X_{k,t})$ is drawn from $\nu_k$ independently from other rewards streams, but we don't assume the rewards are of an i.i.d manner from the corresponding arm. Following Carpentier & Valko (2014) and Bhatt et al. (2023), we give

**Definition 1** (Vanishing Extreme Regret). Let $\pi$ be a policy allocating arm trails, it selects an arm $I_t$ using past observations at time $t$, the *extreme regret* of $\pi$ is

$$\mathcal{R}_T^\pi = \max_{k \leq K} \mathbb{E}[\max_{t \leq T} X_{k,t}] - \mathbb{E}_\pi[\max_{t \leq T} X_{I_t,t}].$$

We say that $\pi$ has a vanishing regret in the *weak* sense if

$$\frac{\mathbb{E}_\pi[\max_{t \leq T} X_{I_t,t}]}{\max_{k \leq K} \mathbb{E}[\max_{t \leq T} X_{k,t}]} \to 1, \text{ as } t \to \infty$$

and $\pi$ has a vanishing regret in the *strong* sense if

$$\max_{k \leq K} \mathbb{E}[\max_{t \leq T} X_{k,t}] - \mathbb{E}_\pi[\max_{t \leq T} X_{I_t,t}] \to 0, \text{ as } t \to \infty.$$

Vanishing extreme regret has been applied as common analytical performance measure since Cicirello & Smith (2005) and Carpentier & Valko (2014), it's trivially achieved asymptotically for bounded distributions using just fully randomized policy. This notion of regret is meaningful in Carpentier & Valko (2014) and Bhatt et al. (2023) as they assume the distributions have unbounded support with the only restriction of finite mean. However, for any unbounded rewards stream $(X_{k,t})$ without finite mean guarantee, we can transform it into bounded variables with finite mean by monotone transformations such as $((\exp X_{k,t}+1)^{-1} \in (0,1))$. Our algorithm doesn't assume finite mean and is based in such monotonic transformations when tackling with unbounded rewards stream, so we propose a novel non-trivial performance measure called *asymptotic optimality*.

**Definition 2** (Asymptotic optimality). Let $\pi$ be a policy selecting an arm $I_t$ at time $t$, we say that $\pi$ is asymptotically optimal in the *weak* sense if

$$\lim_{T \to \infty} \frac{\sum_{t=1}^{T} \delta_{I_t k^*}}{\sum_{t=0}^{T} \delta_{I_t k}} > 1$$

and in the *strong* sense if

$$\lim_{T \to \infty} \frac{\sum_{t=1}^{T} \delta_{I_t k^*}}{\sum_{t=0}^{T} \delta_{I_t k}} = \infty$$

for each $k \neq k^*$,

where $\delta_{ij}$ is Kronecker symbol with

$$i \neq j, \delta_{ij} = 0 \ \& \ i = j, \delta_{ij} = 1$$

and the *asymptotically dominating* arm $\nu_{k^*}$ is defined following Bhatt et al. (2023) as

**Definition 3** (Asymptotically Dominating Arm). For each $k \neq k^*$, if

$$\lim_{T \to \infty} \inf \frac{\mathbb{E}[\max_{t \leq T} X_{k^*,t}]}{\mathbb{E}[\max_{t \leq T} X_{k,t}]} > 1,$$

we say that arm $\nu_{k^*}$ (asymptotically) dominates arm $\nu_k$, writing it $\nu_{k^*} \succ \nu_k$. Also, a stronger version of domination is available that

$$\lim_{T \to \infty} \inf \frac{\mathbb{E}[\max_{t \leq T} X_{k^*,t}]}{\mathbb{E}[\max_{t \leq T} X_{k,t}]} = \infty.$$

## 3 AEB ALGORITHM

In this section, we provide *A\* Extreme Bandit* (AEB) algorithm for extreme bandits. The algorithm is straight-forward and simple to implement, requiring no knowledge of the reward distributions, assuming only that the reward values are real[1]. Its name was inspired by the classic A\* search algorithm for it's analyzed in a similar way to A\* search algorithm (see Section 4).

When designing algorithms, we are motivated by one vision: for better flexibility, the algorithm should produce consistent behavior for any monotone transformation of the reward values. Optimization algorithms with this property are said to support objective functions on manifolds. This vision inspired the development of a effort-based surrogate for the reward values.

We denote the global best reward value received after $t$ total trials by $X_t^*$, which grows over $t$. An expected larger effort of trials is needed for a better $X_t^*$. With this intuition, we provide a surrogate for any reward value $X_{k,t}$.

**Definition 4** (Effort-based Surrogate). For any reward value $X_{k,t}$, if there exists an $n$ such that

$$\forall t < n, X_t^* < X_{k,t} \ \& \ \forall t \geq n, X_t^* \geq X_{k,t},$$

then the *effort-based surrogate* for $X_{k,t}$ is $\text{fn}(X_{k,t}) := n$.

---

[1] However, AEB algorithm can also be applied to problems with non-real reward values if an order-preserving mapping from the reward values to real numbers is provided.

The idea of effort-based surrogate is hard to implement for unbounded reward, but after a proper bounded monotone transformation $\mathtt{BMT}(\cdot)$ with range $(0,1)$, we can transform any unbounded reward $X_{k,t}$ into a $(0,1)$-bounded variable. We use an array $\boldsymbol{a}$ of length $N$ to store the surrogates, and for better discrimination between close reward values, an interpolation process is introduced, hence we give our algorithms for constructing and getting effort-based surrogates $\mathtt{fn}(\cdot)$.

---

**Algorithm 1:** Insert-step after each arm pull to construct $\mathtt{fn}(\cdot)$

---

**Set parameter:** Length of the storage array $N$     **Static initialization:** $\boldsymbol{a} \leftarrow \{0\}$
**Input:** Reward $X_{I_t,t}$ obtained from each arm pull     `// function insert(·)`
Compute index $i = \lfloor N * \mathtt{BMT}(X_{I_t,t}) \rfloor$
**if** $\boldsymbol{a}_i = 0$ **then**
     **if** $X_{I_t,t}$ *is the best reward obtained up to* $t$ **then**
        $\boldsymbol{a}_i \leftarrow t$
     **else if** $X_{I_t,t}$ *is the worst reward obtained up to* $t$ **then**
        Let $\boldsymbol{a}_j$ be the nearest nonzero element to $\boldsymbol{a}_i$ in array $\boldsymbol{a}$
        $\boldsymbol{a}_i \leftarrow \boldsymbol{a}_j/2$
     **else**
        Let $\boldsymbol{a}_j$ and $\boldsymbol{a}_k$ be the nearest nonzero elements on either side to $\boldsymbol{a}_i$ in array $\boldsymbol{a}$
        $\boldsymbol{a}_i \leftarrow (\boldsymbol{a}_j + \boldsymbol{a}_k)/2$

---

**Algorithm 2:** Getting $\mathtt{fn}(\cdot)$ for a given reward value

---

**Input:** $X_{k,t}$                                    `// function get_fn(·)`
Compute index $i = \lfloor N * \mathtt{BMT}(X_{I_t,t}) \rfloor$
**Return:** $\boldsymbol{a}_i$

---

Using $\mathtt{insert}(\cdot)$ function defined by Algorithm 1 and $\mathtt{get\_fn}(\cdot)$ function defined by Algorithm 2, we give our main algorithm AEB (Algorithm 3), where $K$ denotes the number of arms, $I_t \in \mathbb{K}$ denotes the arm chosen at $t$, $N_{k,t} = \sum_{\tau=1}^{t} \delta_{I_\tau k}$ denotes the number of the $k^{th}$ arm pulls up to $t$, $T$ denotes the play horizon, $X_{k,t}^*$ denotes the best reward obtained of the $k^{th}$ arm up to $t$.

---

**Algorithm 3:** A* Extreme Bandit (AEB)

---

**Input:** $\mathbb{K}, T$                **Set parameter:** Relaxation factor $\alpha$
**for** $t = 1 : K$ **do**
     $I_t = t$, pull arm $I_t$, then $\mathtt{insert}(X_{I_t,t})$            `// (pull each arm once)`
**for** $t = K + 1 : T$ **do**
     $I_t = \arg\max_{k \in \mathbb{K}} \ \log \mathtt{get\_fn}(X_{k,t-1}^*) - \alpha * \log N_{k,t-1}$
     Pull arm $I_t$, then $\mathtt{insert}(X_{I_t,t})$

---

**Implementation Notes** For AEB algorithm, we have three parameters that need to be specified in advance, $\mathtt{BMT}(\cdot)$ - the monotone negative transformation with range $(0,1)$, $N$ - length of the storage array, and $\alpha$ - the relaxation factor. With $N$ sufficiently large, despite $\mathtt{BMT}(\cdot)$, the index $i = \lfloor N * \mathtt{BMT}(X_{I_t,t}) \rfloor$ of each $X_{k,t}$ would be well separated, hence the specific value of $N$ won't affect the performance of the main algorithm. We will illustrate this empirically in Section 5.4. Therefore, the only non-trivial parameter of AEB algorithm should be $\alpha$ - the relaxation factor. The relaxation factor is associated with the asymptotic property of the algorithm, which will be discussed in the following section.

## 4   ANALYSIS OF AEB

Recall that $\mathtt{fn}(X_{k,t})$ is the effort-based surrogate for reward $X_{k,t}$ and $N_{k,t}$ denotes the number of the $k^{th}$ arm pulls up to $t$. We start with two intuitive properties of the surrogate below.

**Lemma 1.** *The effort-based surrogate* $\mathtt{fn}(X)$ *is a time-invariant non-decreasing function on reward value $X$.*

This lemma holds naturally as the surrogate of $X$ is constructed by Algorithm 1 after the first reward value with the same index $i = \lfloor N * \mathtt{BMT}(X) \rfloor$ to $X$ is obtained and will never be altered afterwards.

**Lemma 2** (Upper bound on $\mathtt{fn}(X_{k,t})$)**.** *For any reward $X_{k,t}$ obtained from any arm $\nu_k$ and any time $t$,*

$$\mathtt{fn}(X_{k,t}) \le t$$

*Proof.* Suppose that $\mathtt{fn}(X_t^*) \le t$, if $X_{t+1}^* > X_\tau^*$ for all $\tau < t+1$, by definition, we have $\mathtt{fn}(X_{t+1}^*) = t+1$. Else-wise, $X_{t+1}^* \le X_t^*$ then $\mathtt{fn}(X_{t+1}^*) \le \mathtt{fn}(X_t^*) \le t$. Therefore $\mathtt{fn}(X_{t+1}^*) \le t+1$ holds. Given that $\mathtt{fn}(X_1^*) = 1$, inductively we have $\mathtt{fn}(X_t^*) \le t$ for all $t \ge 1$.

Note that $X_{k,t} \le X_t^*$ and by Lemma 1, $\mathtt{fn}(X_{k,t})$ is upper-bounded by $t$. $\qquad\square$

This upper bound leads to the following proposition.

**Proposition 1.** *If the bandit is played by policy* $\mathtt{AEB}$ *with the relaxation factor $\alpha > 1$, selecting an arm $I_t$ at time $t$, then for any arm $\nu_k$, there exists a constant $C_k$ such that*

$$\lim_{T\to\infty} \sum_{t=1}^{T} \delta_{I_t k}/T > C_k.$$

*Proof.* We consider the opposite, hypothesize that there is a set of arms $\{\nu_p\}$ that for $\nu_k \in \{\nu_p\}$, $\lim_{T\to\infty} \sum_{t=1}^{T} \delta_{I_t k}/T > C_k$ and the other set of arms $\{\nu_q\}$ with $\lim_{T\to\infty} \sum_{t=1}^{T} \delta_{I_t k}/T = 0$ for $\nu_k \in \{\nu_q\}$. By Algorithm 3, the policy pulls the arm at time $t$ that maximizes

$$f_{k,t-1} = \log \mathtt{fn}(X_{k,t-1}^*) - \alpha \log N_{k,t-1}, \text{ where } N_{k,t-1} = \sum_{\tau=1}^{t-1} \delta_{I_\tau k}.$$

For $\nu_k \in \{\nu_p\}$ at time $T$, applying Lemma 2, we have

$$f_{k,T} < \log T - \alpha \log C_k T = (1-\alpha) \log T - \alpha \log C_k$$

and $\lim_{T\to\infty} (1-\alpha) \log T = -\infty$ as $\alpha > 1$, so $\lim_{T\to\infty} f_{k,T} = -\infty$. While for $\nu_k \in \{\nu_q\}$, $\lim_{T\to\infty} \sum_{t=1}^{T} \delta_{I_t k}$ is finite, so $\lim_{T\to\infty} f_{k,T} > -\infty$. However, arms with larger $f_{k,T}$ should be preferred by Algorithm 3, leading to a contradiction to our hypothesis. $\qquad\square$

Proposition 1 shows that the number of times each arm is pulled over $T$ is of the same order $\mathcal{O}(T)$. We then provide the main analysis result of $\mathtt{AEB}$.

**Theorem 1.** $\mathtt{AEB}$ *is asymptotically optimal in the weak sense if the relaxation factor $\alpha$ is admissible, i.e., $\alpha > 1$.*

*Proof.* We consider a bandit that has a dominating arm denoted by $k^*$: $\nu_{k^*} \succ \nu_k$ for all $k \ne k^*$,

$$\lim_{T\to\infty} \inf \frac{\mathbb{E}[\max_{t\le T} X_{k^*,t}]}{\mathbb{E}[\max_{t\le T} X_{k,t}]} > 1.$$

As $\lim_{T\to\infty} \sum_{t=1}^{T} \delta_{I_t k} > C_k T = \infty$, after the bounded monotone transformation $\mathtt{BMT}(\cdot)$, without a finite bound assumption on $\mathbb{E}[\max_{t\le T} X_{k,t}]$, the best reward obtained

$$\lim_{T\to\infty} \mathtt{BMT}(X_{k,T}^*) = \mathtt{BMT}(\lim_{T\to\infty} \mathbb{E}[\max_{t\le T} X_{k,t}])$$

hence

$$\lim_{T\to\infty} \mathtt{BMT}(X_{k^*,T}^*)/\mathtt{BMT}(X_{k,T}^*) > 1.$$

With $N$ sufficiently large to discriminate the index of $\mathtt{BMT}(X_{k^*,T}^*)$ and $\mathtt{BMT}(X_{k,T}^*)$, we have

$$\lim_{T\to\infty} \mathtt{fn}(X_{k^*,T}^*)/\mathtt{fn}(X_{k,T}^*) > 1.$$

By proposition 1 and its proof, we know that when the policy keeps pulling the same set of arms, the $f_{k,T}$ values of this set of arms will decrease asymptotically for $\alpha > 1$, thus giving the other set of arms a chance to be pulled. So in an asymptotic sense, the $f_{k,T}$ values of all arms are equal, leading to

$$\lim_{T \to \infty} \frac{\log \mathtt{fn}(X^*_{k*,T}) - \alpha \log N_{k*,T}}{\log \mathtt{fn}(X^*_{k,T}) - \alpha \log N_{k,T}} = 1.$$

With the above two results we can get

$$\lim_{T \to \infty} \frac{N_{k*,T}}{N_{k,T}} = \lim_{T \to \infty} \frac{\sum_{t=1}^{T} \delta_{I_t k^*}}{\sum_{t=0}^{T} \delta_{I_t k}} > 1.$$

$\square$

However, if $N$ is too small to discriminate the index of $\mathtt{BMT}(X^*_{k*,T})$ and $\mathtt{BMT}(X^*_{k,T})$, we have $\lim_{T \to \infty} N_{k*,T}/N_{k,T} \geq 1$. The asymptotic optimality of $\mathtt{AEB}$ is albeit a weak one, Nishihara et al. (2016) proved that no policy can asymptotically achieve no extreme regret, i.e., no policy is asymptotically optimal in the strong sense.

**Comparison with the A\* search algorithm**  A\* (pronounced "A-star") is a graph traversal and path search algorithm used to find the shortest path from a specified source to a specified goal. It is widely used in computer science due to its completeness and optimality (Hart et al., 1968). A\* can be considered an extension of Dijkstra's algorithm, but it achieves better results by using heuristics to guide its search. In each iteration of its main loop, A\* determines the path to extend by minimizing the function

$$f(n) = g(n) + h(n).$$

Here, $n$ represents the next node on the path, $g(n)$ is the cost of the path from the start node to $n$, and $h(n)$ is a heuristic function that estimates the cost of the cheapest path from $n$ to the goal. Once a path is found, A\* terminates and returns the path.

**Theorem 2** (Optimality of A\*). *If the heuristic function is admissible, meaning it never overestimates the actual cost to reach the goal, A\* is guaranteed to return a optimal (least-cost) path from the start to the goal.*

*Proof.* Without loss of generality, we assume that the cost of the optimal path is $C$. As the heuristic function is assumed to be admissible, we know that $f(n^*) \leq C$ for any node $n^*$ on the optimal path. Considering a node $n'$ with $g(n') = C$ on a sub-optimal path with cost $C + d$ $(d > 0)$, then we have $f(n') = C + h(n') > C \geq f(n^*)$. So $n'$ would never be expanded until every node on the optimal path is expanded, i.e., the optimal path is found and returned. $\square$

The early nodes on a sub-optimal path might be expanded, but as the expanding process goes on, the $f(n')$ of a new node $n'$ that can be expanded would be inferior to those of nodes on the optimal path. $\mathtt{AEB}$ works in a similar way to A\*, a sub-dominating arm $\nu_k$ might be pulled many times, but with $\alpha > 1$, $f_{k,T}$ of would go inferior to $f_{k^*,T}$, leading to more pulls on the dominating arm $\nu_{k^*}$. However, $\mathtt{AEB}$ doesn't terminate like A\* does.

## 5  PRACTICAL PERFORMANCE

In this section, we empirically evaluate $\mathtt{AEB}$ with the different competitors on synthetic data. The competitors include ThresholdAscent (Streeter & Smith, 2006b), ExtremeHunter (Carpentier & Valko, 2014), ExtremeETC (Achab et al., 2017), Max-Median (Bhatt et al., 2023) and QoMax (Baudry et al., 2022). We use parameters suggested in the original papers and the comprehensive work of Baudry et al. (2022). All numerical results in this section are averaged over 1000 independent experiments.

## 5.1 TIME AND STORAGE COMPLEXITY

We present in Table 1 the storage and computational complexity for a time horizon $T$ needed by the different adaptive and ETC (Explore-Then-Commit) algorithms that we examine, using the mentioned parameters. For the complexities of the baselines, we refer to Baudry et al. (2022). The lowest values in each category are highlighted in blue.

For AEB (Algorithm 3), the memory needed is $K + N$, $K$ to store the best reward obtained by each arm, and $N$ for the effort-based surrogate array $\boldsymbol{a}$. The computational complexity of $\log \texttt{get\_fn}(X^*_{k,t-1}) - \alpha * \log N_{k,t-1}$ is $\mathcal{O}(1)$ while the over-all computational complexity of $\texttt{insert}(X_{I_t,t})$ is $\mathcal{O}(N \log T)$. To identify $I_t$ for $t = K+1 : T$, a naive implementation takes $K-1$ comparisons, but a priority queue can be applied to get a reduced $\mathcal{O}(\log K)$ complexity. So the over-all time complexity of AEB for a time horizon $T$ is $\mathcal{O}(T \log K + N \log T)$. However, we can avoid performing $\texttt{insert}(X_{I_t,t})$ operations and safely discard the surrogate for $X_{I_t,t} < \min\{X^*_{k,t}\}$, resulting in a memory complexity of $\mathcal{O}(K)$ and a time complexity of $\mathcal{O}(T \log K)$.

Table 1: Average memory & time complexities and of Extreme Bandit algorithms

| Algorithm | Memory | Time |
|---|---|---|
| ThresholdAscent | $s$-fixed | $\mathcal{O}(KT)$ |
| ExtremeHunter | $T$ | $\mathcal{O}(T^2)$ |
| Max-Median | $T$ | $\mathcal{O}(KT \log T)$ |
| QoMax-SDA | $\mathcal{O}((\log T)^2)$ | $\mathcal{O}(KT \log T)$ |
| **AEB** (ours) | $\mathcal{O}(K)$ | $\mathcal{O}(T \log K)$ |
| ExtremeETC | $\mathcal{O}(K(\log T)^3)$ | $\mathcal{O}(K(\log T)^6)$ |
| QoMax-ETC | $\mathcal{O}(K(\log T)^2)$ | $\mathcal{O}(K(\log T)^3)$ |

AEB presents the lowest storage cost compared to all competitors within our knowledge. It should be noted that ThresholdAscent considers the $s$ best rewards observed so far where $s$ is a parameter of the algorithm, but it lacks theoretical grounding though utilizes a desirable $s$-fixed amount of data. Among all adaptive (non-ETC) algorithms, AEB is the most computationally efficient. However, ETC algorithms first explore for a certain amount of trials and then commit to a single arm for all subsequent pulls, which naturally under-perform when dealing with non-stationary arms.

## 5.2 PERFORMANCE EVALUATION

In Bhatt et al. (2023), two criteria for evaluating the empirical finite sample performance is employed: **(I)** the *extreme regret defined in Definition* 1 and **(II)** the *fraction of beat arm pulls*. Most works report only **(I)**, Baudry et al. (2022) note that estimating the expectation $\mathbb{E}[\max_{t \leq T} X_{I_t,t}]$ required for calculating the extreme regret is difficult. They also mention that approximations of $\mathbb{E}[\max_{t \leq T} X_{k^*,t}]$ are only available for a few specific families and the use of standard Monte-Carlo estimators can lead to high variance for heavy-tailed distributions. In the following sections, we mainly employ criterion **(II)** for evaluation, it's proven to be a robust performance indicator in the experiments and analysis of Bhatt et al. (2023) and Baudry et al. (2022).

## 5.3 NUMERICAL RESULTS

In this section, we illustrate the performance of AEB on the most widely studied distributions inspired by real data, along with newly proposed non-i.i.d distributions. Most of the competitors' results are also reported by Baudry et al. (2022). According to Section 5.4, we set $N = 1 \times 10^5$, $\texttt{BMT}(x) = x^{-1}$ for $x \in [1, \infty)$ and $\texttt{BMT}(x) = e^{-x}$ for $x \in [0, \infty)$, and $\alpha = 1$.

1. *Pareto Arms*: The tail distribution $\bar{F}(x) \sim a_k x^{-\lambda_k}$ for $k = 1, 2, \cdots, K$. This is motivated by heavy-tailed data. We choose $K = 7$ Pareto distributions with $\lambda_k \in [2.5, 2.8, 4, 3, 1.4, 1.4, 1.9]$. All arms have a scaling $a_k = 1$ except $\nu_5$ with $a_5 = 1.1$. Hence $\nu_5$ is the dominating arm from a slight margin. The results are illustrated in the left of Figure 1.

2. *Exponential Arms*: The tail distribution $\bar{F}(x) \sim a_k e^{-\lambda_k x}$ for $k = 1, 2, \cdots, K$. This is motivated by exponential-tailed data. We choose $K = 10$ Exponential arms with parameters $\lambda_k \in [2.1, 2.4, 1.9, 1.3, 1.1, 2.9, 1.5, 2.2, 2.6, 1.4]$. The results are shown in the right Figure 1.

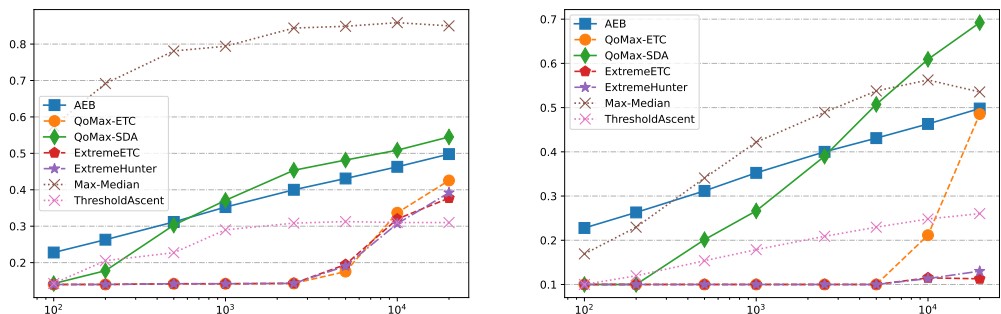

Figure 1: Fraction of best arm pulls for two kinds of distributions for $T \in \{100, 200, 500, 1 \times 10^3, 2.5 \times 10^3, 5 \times 10^3, 1 \times 10^4, 2 \times 10^4\}$

### 5.4 Effects of parameter selection

In this section, we demonstrate the impact of selecting the length $N$ of the storage array $a$ and the relaxation factor $\alpha$, based on exp.1 in Bhatt et al. (2023) for its simplicity: $K = 5$ Pareto distributions with tail parameters $\lambda_k \in [2.1, 2.3, 1.3, 1.1, 1.9]$. We take $\text{BMT}(x) = x^{-1}$ as $x \in [1, \infty)$ from the Pareto distributions. The results are shown in Figure 2.

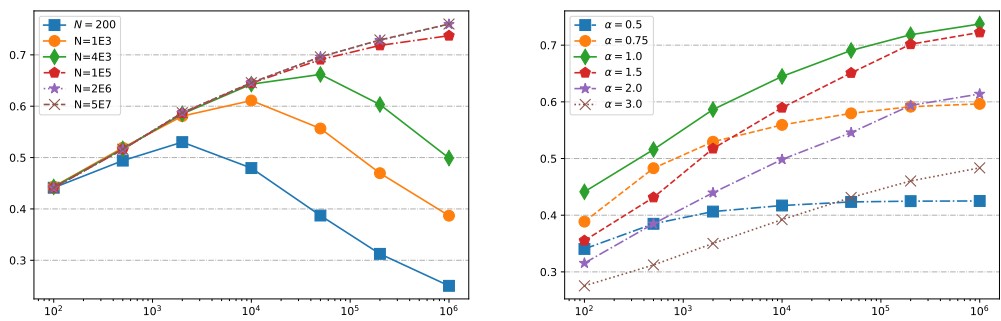

Figure 2: Fraction of best arm pulls using different parameters for $T \in \{100, 500, 2 \times 10^3, 1 \times 10^4, 5 \times 10^4, 2 \times 10^5, 1 \times 10^6\}$

The numerical results show that the performance of the algorithm is positively correlated with the value of $N$. When $N$ is much smaller than $T$, it is difficult to discriminate the best rewards among arms, so the performance of the algorithm tends to approach a simple random policy. When $N$ approaches $T$, the further increase of $N$ brings marginal improvement. When $\text{BMT}(x)$ has a relatively uniform distribution, we suggest that setting $N$ to the same order as $T$ would be sufficient. Setting $N$ too small will result in a loss of asymptotic performance, despite higher efficiency. In the experiment investigating the influence of $\alpha$, we set $N = 1 \times 10^5$. Although the algorithm has asymptotic guarantees when $\alpha > 1$, empirical results show that the convergence speed slows down significantly. When $\alpha < 1$, the algorithm is easily misled by sub-optimal arms. Therefore, we recommend setting $\alpha$ equal to or slightly larger than 1.

**Take-away parameter suggestions:** Set array length $N \approx T$, relaxation factor $\alpha \gtrsim 1$.

## 6 CONCLUSION

We provided a universal Extreme Bandit algorithm `AEB` with the minimal memory complexity and time complexity (in non-ETC algorithms). `AEB` is the first algorithm tackling with non-stationary rewards, which offer a more precise representation of real-world data sources. The universality of the algorithm is achieved by surrogating reward $X$ with the effort taken to find the first reward as good as $X$. For non-i.i.d rewards without bound on the finite mean, we first proposed asymptotic optimality for analysis and we established that the best arm will have the most allocated trials eventually by `AEB`. Numerical results in recognized experimental settings confirmed the efficiency and finite-sample performance of `AEB`.

An interesting direction is to establish vanishing regret for `AEB` under stronger assumptions extensively studied in the Extreme Bandit literature, although it currently doesn't seem to be beneficial in enhancing the performance of `AEB` on diverse application scenarios.

This work is motivated by black-box optimization (BBO) problems, for example, hyper-paremeter optimization. Its contribution may inspire more people to integrate more insights from the bandit literature into BBO problems, and the authors are in the process of assessing and improving the proposed algorithm in realistic BBO problems.

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
