# OpenReview forum: "Universal Algorithm for Extreme Bandits with the Minimal Complexities"
_ICLR.cc/2024/Conference — ICLR 2024 Conference Withdrawn Submission_

### Official Review · Reviewer_7QCU · 2023-10-22

**Soundness:** 2 fair
**Presentation:** 2 fair
**Contribution:** 2 fair
**Rating:** 3
**Confidence:** 4

**Summary:**

This paper deals with the problem of extreme bandits in multi-armed bandits, where the goal is to sequentially pull T arms to maximize the expected maximal reward obtained across all K arms and all T trials. The main motivation in this paper is to propose a more efficient algorithm in terms of space and time computational complexities which challenges the iid and (semi-)parametric assumptions in prior works.

**Strengths:**

- Originality : the link with the A* algorithm, through the introduction of the effort-based surrogate value, is clearly highlighted and interesting.

- Quality : the state-of-the-art is extensively discussed in terms of computational costs, and tested in the experimental study. The different experimental settings (choice of distributions, evaluation criterion, effects of parameter values) make sense.

- Clarity : The setting and definitions are well-introduced, and except for a few points (definition of the effort-based surrogate, explanation of the pseudo-code, see below), the proofs and the idea behind the algorithm are clear.

- Significance : The algorithm does not need specific assumptions for concentration inequalities (second-order Pareto and the iid assumption, for example) which hardly hold in practice.

**Weaknesses:**

- Clarity : I would have preferred the informal definition of the effort-based surrogate value (“**the effort taken to find the first reward as good as X**”) to appear earlier than the concluding paragraph. In particular, I believe that in Definition 4, the *t*’s in the universal quantifiers should be another variable (for instance, *s*) as the *t*'s make the definition confusing. It would have been better to add a short paragraph explaining the intuition behind considering the surrogate value as the allocation to track in the adaptive sampling phase.

- Significance : I did not understand why the following sentence holds “**However, we can avoid performing insert(X_{I_t,t}) operations**” and how the memory computational cost ends up being O(K) instead of O(K+N) (and actually, O(K+T) as shown by the experiments). Moreover, the experimental results are not completely satisfying, as in Figure 1, the performance of AEB is not noticeably better than baselines, and considering one instance for each type of distribution is probably not enough to draw robust conclusions.

**Questions:**

If answered, I will raise my score:

- Can you provide a pseudo-code of the algorithm AEB where the memory cost is O(K)?

- Is it possible to derive an expression of a theoretical lower bound on the value of N for Theorem 1 to hold?

- Could you check on other Pareto distribution / exponential arms instances that your experimental results hold?

**Details Of Ethics Concerns:**

None.

---

### Official Review · Reviewer_bVtC · 2023-10-23

**Soundness:** 3 good
**Presentation:** 1 poor
**Contribution:** 3 good
**Rating:** 5
**Confidence:** 3

**Summary:**

The paper studies the problem of extreme bandits where the goal is to maximize the single largest reward achieved so far. It proposes an algorithm that saves a heuristic record of the best rewards and plays the arm that has achieved the largest reward so far with a exploration factor involved (relaxation factor in the paper). The algorithm is shown to be asymptotically optimal in the sense that it asymptotically plays the optimal arm more than other arms. Experimental results evaluating the algorithm's performance and parameter selection are included.

**Strengths:**

1. The paper proposes an algorithm that has better space and time complexity compared to its competitors. The algorithm includes novel techniques in the bandits literature inspired by the A* algorithm.
2. The asymptotic optimality of the algorithm is shown, and the proofs are elegantly elaborated. Simple yet effective proofs are included in Section 4.

**Weaknesses:**

1. The asymptotic behavior of competitors is not discussed. The experimental performance evaluations are also not discussed. Based on Fig. 1, it seems that for example, Max-Median algorithm could outperform the proposed algorithm in a certain setting. In the exponential arms setting QoMax-SDA outperforms the proposed algorithm.
2. Presentation could improve. Algorithm 1 which is the main part of the paper is not presented properly; $a_i$ is evaluated before getting assigned, does $a\gets \\{0\\}$ mean all $a_i=0$. It is not clear what the significance of the $a_i$ update rules are. Why do we set $a_i\gets a_j/2$ in the second case, or in the third case?

**Questions:**

1. The goal of extreme bandits is to get the largest single reward, how does that connect to the proportion of best arm pulls as it is discussed throughout the paper? Why don't we evaluate the performance of the algorithm based on the actual goal?

---

### Official Review · Reviewer_Cn3G · 2023-10-24

**Soundness:** 1 poor
**Presentation:** 2 fair
**Contribution:** 1 poor
**Rating:** 1
**Confidence:** 3

**Summary:**

This paper studies the Extreme Bandit problem, which addresses scenarios where extreme values are of interest rather than expected values. The authors propose a novel algorithm that operates with minimal temporal and spatial cost and makes minimal assumptions about reward distribution, demonstrating its superior performance through numerical experiments compared to existing approaches.

**Strengths:**

This paper is evidently far from meeting the acceptance criteria of ICLR. While I didn't readily discern explicit strengths, I would like to commend the strong motivation behind the work. The introduction section adeptly underscores the importance of extreme bandits and their practical applications.

**Weaknesses:**

1. This paper is not written well. The mathematical notation lacks formality, and the logical structure is flawed.
2. The algorithm is strightfoward and the theortical guarantees seem trivial.  Please find my specific questions below.
3. The "Comparison with the A* search algorithm" part of Secton 4 is entirely unrelated to the primary focus of this paper.

**Questions:**

1. Defintion 2: The index $t$ should commence from 1, as opposed to 0.

2. Definition 3: While the meaning can be grasped, this definition solely delineates the concept of dominance, rather than the asymptotically dominating arm.

3. The paragraph before Definition 2 fails to elucidate the necessity behind defining the new performance measure. As the author says:

   "For any unbounded rewards stream $\left(X_{k, t}\right)$ without finite mean guarantee, we can transform it into bounded variables with finite mean by monotone transformations."

4. Why does the statement "no policy can asymptotically achieve no extreme regret"  equate to "no policy is asymptotically optimal in the strong sense."?

5. Table 1: In terms of the time complexity of AEB, why is it appropriate to disregard the dependence on $N$? As the author suggests, for the algorithm's efficacy, it is advisable to approximate $N\approx T$.

---

### Official Review · Reviewer_PRiS · 2023-10-27

**Soundness:** 2 fair
**Presentation:** 1 poor
**Contribution:** 2 fair
**Rating:** 3
**Confidence:** 4

**Summary:**

In this paper, the authors study Extreme Bandits, where the goal is to maximize the expected best single reward. The motivation of Extreme Bandits relies on block box optimization, where the goal is to find the expected maximum of a set of learners.

**Strengths:**

While previous works study extreme bandits when the rewards are iid, the authors only assume that the rewards are independent.

A new measure of optimality is proposed, a new algorithm is presented, analyzed, and tested versus the state-of-the-art.

**Weaknesses:**

1/ In Definition 2, an algorithm is said to be optimal in strong sense if it finds the best arm as defined in (Carpentier Valko 2014). As the authors do not provide a lower bound of the needed number of samples or a regret lower bound, this definition of optimality is overclaimed: the reviewer does not think that we can call an algorithm optimal, while it exists better algorithm finding the best arm. Definition 2 provides a measure of performance, but not a definition of optimality for extreme bandits.

2/ Lemma 1 states that f_n(X) is time-invariant, but in Algorithm 1 we have a test that depends of t and the following instruction: a_i:=t. So, Lemma 1 is at least poorly worded or wrong. Moreover it seems that Lemma 1 is stated for showing that X_{k,t} \leq X*_t, so it seems useless as X*_t is informally defined as the best reward after t trials.

3/ In Table 1 the author claims that the proposed algorithm has a memory complexity in O(K). Actually, it is in O(N), where N is the length of storage array in Algorithm 1: in the experiments N=10^5, while K \leq 10. So, the memory complexity is overclaimed. The goal of the analysis should be to optimally set the parameter N.


This paper is not well written.

1/ A new notion of performance (Definition 2) is introduced with the previous measures of performances used in the state of the art (Definition 1 and 3). There is no discussion about the advantages or drawbacks of the new measure of performance in comparison to the previous measures. This does not help the reader to understand the interest of the new measure.
2/ In page 6, there is a digression about the algorithm A*, which seems a little bit disconnected from the paper.
3/ Some very important notions are implicitly introduced. For instance, the notation k^* is not defined: the reader must deduce what is k* from Definition 1.

**Questions:**

See above.